# Multi Spherical Wave Imaging Method Based on Ultrasonic Array

**DOI:** 10.3390/s22186800

**Published:** 2022-09-08

**Authors:** Zhi-Ying Liu, Ping Zhang, Bi-Xing Zhang, Wen Wang

**Affiliations:** 1Key Laboratory of Acoustics, Institute of Acoustics, Chinese Academy of Sciences, Beijing 100190, China; 2University of Chinese Academy of Sciences, Beijing 100049, China; 3Key Laboratory of Nondestructive Testing, Ministry of Education, Nanchang Hangkong University, Nanchang 330063, China

**Keywords:** multi spherical wave imaging, plane wave imaging, composite spherical wave imaging, ultrasonic array

## Abstract

The imaging range of traditional plane wave imaging is usually limited by the directivity of the plane wave. In this paper, a multi spherical wave imaging method based on an ultrasonic array is proposed, which radiates both compression and shear waves in a solid medium to form the multi spherical wave. Firstly, excitation characteristics of the multi spherical wave are analyzed theoretically and the calculation method of echo delay of multi spherical wave imaging is derived. Multi spherical wave imaging is compared with conventional ultrasonic plane wave imaging by designing experiments. Compared with ultrasonic plane wave imaging, multi spherical wave imaging is not limited to the size of the transducer and can greatly improve the detection range. In addition, compared with the multi plane wave imaging method, the multi spherical wave imaging algorithm is relatively simple, fast, and has high application value.

## 1. Introduction

At present, with the improvement of testing quality and real-time capability standards in the field of industrial nondestructive testing, the requirements of ultrasonic imaging technology are becoming higher and higher [1]. In the 1970s, research on high-speed ultrasound imaging began gradually. In 1979, Delannoy et al. [2,3] first proposed the method of parallel processing to make a single sound wave generate the whole scan image. The experimental results show that parallel processing imaging can generate 1000 frames per second, and each frame contains 70 scan lines. In 1984, Shattuck et al. [4] realized a parallel processing scheme by simultaneously collecting four b-scan mode image data from each individually broadened transmitting pulse, thus increasing the original data acquisition rate by four times. This parallel processing scheme shows that an echo generated from a single transmitted pulse can image the entire fault plane on the premise that a single pulse can completely radiate the region of interest. It was not until 15 years later that Fink et al. [5,6,7] successfully applied the concept of plane wave radiation to achieve ultra-high-speed imaging of 5000 frames per second. In 2009, Montaldo et al. [8] proposed that, combined with the principle of ultrasonic composite imaging, composite plane wave imaging can achieve high-quality ultrasonic imaging without reducing the ability of the high frame rate. Later, Denarie et al. [9] proposed a motion compensation technique based on cross-correlation, which effectively improved the signal-to-noise ratio and contrast ratio of composite plane wave imaging in physiological tissues.

All the previous research on plane wave imaging algorithms were carried out around a single longitudinal wave. However, in recent years, the fusion of multiple imaging methods and multi-wave imaging is an important development trend in the field of acoustic imaging. Among them, representative imaging methods include multi-mode full-focus imaging [10], multi-wave focusing and imaging [11], and multi-wave and multi-component exploration [12], etc.

This paper mainly focuses on fast ultrasonic multi spherical wave imaging based on ultrasonic array. Firstly, the acoustic emission and imaging principles about the plane wave are reviewed and the relevant experiments are carried out. The multi spherical wave imaging method using ultrasonic array is proposed based on the idea of plane wave imaging. The implementation process of spherical wave emission is analyzed and the corresponding imaging algorithm is established. Then, based on the idea of ultrasonic multi wave focusing, the composite multi-spherical wave emission and imaging detection is studied, and the algorithm and calculation basis are obtained. Finally, the verification experiments with correlation analysis about imaging quality and speed are carried out.

## 2. Theoretical Analysis of Traditional Imaging

### 2.1. Principle of Single Plane Wave Imaging

The conventional single plane wave imaging mode controls all the array elements of the ultrasonic transducer to simultaneously apply non-delay pulse excitation [13], as shown in Figure 1. At this time, a plane wave front will form inside the solid medium, and the whole region corresponding to the size of the transducer can be scanned in parallel. Then, the ultrasonic echo data received by the transducer array is processed by signal processing, usually time delay processing, to obtain a frame image of ultrasonic plane wave detection.

As shown in Figure 1, the *x*-axis direction is parallel to the ultrasonic phased array, the *z*-axis represents the direction of the imaging depth, and the plane wave is excited by the phased array for scanning. According to the above plane wave emission mode and the geometric relation, the propagation path $d_{p}$ of the plane wave transmitted to point (*x*, *z*) and then reflected to the *n*-th element of the phased array can be expressed as [14]:(1)$$d_{p}=z+\sqrt{{\left(x-x_{n}\right)}^{2}+z^{2}}$$
where $x_{n}$ represents the abscissas of the *n*-th element.

Assuming that the array center of the ultrasonic array is taken as the reference, the delay between the echo signals at the *n*-th element and the array center is:(2)$$\tau_{n}=\frac{d_{p}-\left(z+\sqrt{x^{2}+z^{2}}\right)}{c}=\frac{\sqrt{{\left(x-x_{n}\right)}^{2}+z^{2}}-\sqrt{x^{2}+z^{2}}}{c}$$
where *c* represents the compressional wave velocity for single plane wave imaging. For the imaging point (*x*, *z*), we conduct beam synthesis of signals received by all array elements according to the formula below.
(3)$$S^{PW}\left(t\right)=\sum_{n=1}^{N}S_{n}\left(t-\tau_{n}\right)$$
where $S_{n}(t)$ is the echo signal received by the *n*-th element. It shows that the beam synthesized according to Equation (3) can be obtained for each imaging point (*x*, *z*). Then, the reconstructed image of single plane wave emission can be obtained. Meanwhile, combined with the plane wave emission mode in Figure 1, it can be found that the effective area of a single plane wave scan is seriously affected by the aperture of the ultrasonic array, and the effective radiation range of the plane wave is only the area directly below the transducer. Therefore, the effective area of single plane wave imaging is shown in Figure 2.

As can be seen from the figure above, the single plane wave imaging region is greatly limited. The imaging region in the horizontal direction is mainly affected.

### 2.2. Principle of Coherent Plane Wave Compounding

In plane wave imaging, all elements of the transducer array are launched at the same time and produce a vertical downward plane wave. It is not difficult to find that if the time delays between adjacent arrays are the same, the radiation sound wave will be an inclined plane wave, however, change the delay, it will change the angle of the plane wave.

For single plane wave imaging, the transducer array emits a vertically downward or tilted plane wave, which has poor focusing effect and low imaging contrast and SNR (signal-to-noise ratio) [15,16]. To this end, we consider Figure 3 to improve the above situation by emitting a series of plane waves with different deflection angles. We emit plane waves in different directions, conduct beamforming operation on the echo data, and then conduct coherent superposition processing on the beamforming results of all plane waves in different directions to obtain the final imaging result, so as to improve the resolution and SNR of the detected image. This method is called composite plane-wave imaging [8].

In Figure 3, the *x*-axis represents the direction of the transducer array and *z*-axis represents the direction perpendicular to the transducer array. In the process of composite plane wave imaging, the ultrasonic phased array needs to emit *K* plane waves with different deflection angles.

In each launching process, the excitation delay time required by the *n*-th element can be expressed as:(4)$$\Delta t_{n}=T_{c}+\frac{x_{n}\tan\alpha_{m}}{c}$$
where $T_{c}$ is a time constant, which is used to ensure that the transmission delay is a nonnegative number.

According to the geometric relation of Figure 3, for a given deflection angle $\alpha_{m}$ (*m* = 1, 2, ..., *K*), the propagation distance $d_{p}$ of the plane wave from the transmitting wave front to the point (*x*, *z*) and then reflected to the *n*-th element of the phased array can be expressed as:(5)$$d_{p}=x\sin\alpha_{m}+z\cos\alpha_{m}+\sqrt{{\left(x-x_{n}\right)}^{2}+z^{2}}$$

Additionally, taking the array center of the ultrasonic phased array as the reference, the time delay between the echo signals at the *n*-th element and the array center is:(6)$$\tau_{n}=\frac{d_{p}-\left(x\sin\alpha_{m}+z\cos\alpha_{m}+\sqrt{x^{2}+z^{2}}\right)}{c}=\frac{\sqrt{{\left(x-x_{n}\right)}^{2}+z^{2}}-\sqrt{x^{2}+z^{2}}}{c}$$

It shows that this time delay is independent of the deflection angle $\alpha_{m}$.

The coherent plane wave imaging is obtained by delay superposition (beam synthesized) of the signals received by each array element according to the following formula:(7)$$\left\{\begin{array}{l} S^{CPW}\left(t\right)=\sum_{m=1}^{K}S_{m}^{PW}\left(t\right)\\ S_{m}^{PW}\left(t\right)=\sum_{n=1}^{N}S_{mn}\left(t-\tau_{n}\right) \end{array}\right.$$
where $S_{mn}(t)$ is the echo signal received by the *n*-th element when the plane wave with the deflection angle $\alpha_{m}$ is transmitted. $S^{CPW}\left(t\right)$ and $S_{m}^{PW}\left(t\right)$ are the imaging functions for the coherent plane wave and the *m*-th plane wave, respectively.

In the actual imaging process, we first conduct beam synthesized signals for all imaging points according to the second equation of Equation (7) for each given deflection angle, and *K* reconstruction images corresponding to the different deflection angles are obtained. Then, the coherent plane wave imaging can be obtained by stacking *K* reconstruction images according to the first equation of Equation (7). This method is called composite plane wave imaging.

The composite plane wave imaging is to coherently combine each single plane wave imaging with different deflection directions. Compared with single plane wave imaging, composite plane wave imaging can further improve the imaging quality and maintain a faster imaging speed.

### 2.3. Principle of Spherical Wave Imaging

Spherical wave imaging is put forward based on the launch of the plane wave. The transducer array emits a spherical wave to the detection area by controlling the time delay for each element. This is equivalent to setting a spherical center of the spherical wave on the other side of the detection area, and using the distance between the spherical center and each element to calculate the time delay between each array element. The ultrasonic wave generated under this time delay rule forms a spherical wave and can scan the detection area.

#### 2.3.1. Single Spherical Wave Imaging

Figure 4 shows the sound propagation model of single spherical wave imaging. Take the coordinate system (*x*, *z*) of which the origin is located at the transducer center. The spherical center coordinate of the spherical wave is $\left(0,-f_{oc}\right)$. We assume that a virtual point source emits an ultrasonic wave at the center of the sphere. The propagation time when the signal reaches each element and the delay between adjacent elements can be obtained according to the geometric distance between the spherical center and each element. Such delay rules are loaded onto each transducer element, and a spherical wave generates and radiates outward from the center of the virtual point source, as shown in Figure 4. It should be noted that the virtual point source does not exist, and the purpose of setting the virtual point source is to get the delay rule of the spherical wave emitted by the transducer array.

In the figure above, the spherical wave emitted by the transducer array is equivalent to radiation from the virtual point source place $\left(0,-f_{oc}\right)$. In the launching process, the excitation delay time required by the *n*-th matrix element can be expressed as:(8)$$\Delta t_{n}=T_{c}+\frac{f_{oc}-\sqrt{f_{oc}^{2}+x_{n}{}^{2}}}{c}$$

According to geometric relation, the ultrasonic wave propagation distance $d_{oc}$ from the virtual point source to point (*x*, *z*) and then reflected to the *n*-th element of the phased array can be expressed as:(9)$$d_{oc}=\sqrt{x^{2}+{\left(z+f_{oc}\right)}^{2}}+\sqrt{{\left(x-x_{n}\right)}^{2}+z^{2}}$$

Therefore, the time delay of scattering echo between the *n*-th element and the array center is:(10)$$\tau_{n}=\frac{d_{oc}-\left(\sqrt{x^{2}+z^{2}}+\sqrt{x^{2}+{\left(z+f_{oc}\right)}^{2}}\right)}{c}=\frac{\sqrt{{\left(x-x_{n}\right)}^{2}+z^{2}}-\sqrt{x^{2}+z^{2}}}{c}$$

It can be found that the time delay in Equation (10) for spherical wave imaging is equal to that in Equation (6) for plane wave imaging.

Then, the imaging function for spherical wave imaging can be expressed as:(11)$$S^{SW}\left(t\right)=\sum_{n=1}^{N}S_{n}\left(t-\tau_{n}\right)$$
where $S_{n}(t)$ is the echo signal received by the *n*-th element under the spherical wave emission mode.

By changing the position of the imaging points in turn, the scattering echo signal of each imaging point can be beam synthesized according to Equation (11) to obtain the amplitude information of the point. Then, the ultrasonic reconstruction image based on the ultrasonic phased array with the virtual point source can be obtained and is shown in Figure 5.

It can be seen from Figure 5 that the imaging region of spherical waves is less affected by the aperture of the transducer. Compared with Figure 2, the effective imaging range is significantly improved. The introduction of virtual point source lays a foundation for ultrasonic multi spherical wave imaging.

Ultrasonic phased array controls the shape and direction of the acoustic beam by the time delay and amplitude of the excitation signal of each array element. The spherical wave method is actually a special phased array method where the acoustic beam is also obtained by the delay and amplitude of the signals of each array element. However, the conventional phased array obtains focused acoustic beams while the spherical wave method obtains divergent spherical waves.

#### 2.3.2. Coherent Spherical Wave Compounding

Similar to single plane wave imaging, single spherical wave imaging has poor focusing effect. Although the imaging range is improved compared with traditional single plane wave imaging, the imaging resolution and SNR are not improved due to the acoustic beam divergence of the single spherical wave. Similarly, the coherent spherical wave method is proposed by the multi spherical wave with different deflection directions. As shown in Figure 6, spherical waves with different deflection directions can be obtained by changing the position of the virtual point source. Coherent spherical wave imaging is the coherent superposition of image functions of the spherical wave with different deflection directions. This method is called composite spherical wave imaging.

In the process of composite spherical wave imaging, the ultrasonic phased array needs to constantly change the virtual point source position to determine the delay rule and obtain the spherical waves with different deflection angles $\alpha_{m}$ (m=1,2,...,K), respectively. Additionally, *K* is the total number of the deflection angles. In each launching process, to form a deflected spherical wave at angle $\alpha_{m}$, the excitation delay time required by the *n*-th matrix element can be expressed as:(12)$$\Delta t_{n}=T_{c}+\frac{f_{oc}-\sqrt{{\left(x_{n}-f_{oc}\sin\alpha_{m}\right)}^{2}+{\left(f_{oc}\cos\alpha_{m}\right)}^{2}}}{c}$$

According to the geometric relation, when the deflection angle is $\alpha_{m}$, the propagation distance $d_{oc}$ of the spherical wave transmitted from virtual point to point (*x*, *z*) and then reflected to the *n*-th element of the phased array can be expressed as:(13)$$d_{oc}=\sqrt{{\left(x+f_{oc}\sin\alpha_{m}\right)}^{2}+{\left(z+f_{oc}\cos\alpha_{m}\right)}^{2}}+\sqrt{{\left(x-x_{n}\right)}^{2}+z^{2}}$$

It is also assumed that the array center of the ultrasonic phased array is taken as the reference, the time delay between the echo at the *n*-th element and the array center can be rewritten as:(14)$$\begin{array}{cc} \tau_{n} & =\frac{d_{oc}-\left(\sqrt{{\left(x+f_{oc}\sin\alpha_{m}\right)}^{2}+{\left(z+f_{oc}\cos\alpha_{m}\right)}^{2}}+\sqrt{x^{2}+z^{2}}\right)}{c}\\ & =\frac{\sqrt{{\left(x-x_{n}\right)}^{2}+z^{2}}-\sqrt{x^{2}+z^{2}}}{c} \end{array}$$

Then, coherent spherical wave imaging can be obtained by delay superposition (beam synthesized) of the signals received by each array element according to the following formula:(15)$$\left\{\begin{array}{l} S^{CSW}\left(t\right)=\sum_{m=1}^{K}S_{m}^{SW}\left(t\right)\\ S_{m}^{SW}\left(t\right)=\sum_{n=1}^{N}S_{mn}\left(t-\tau_{n}\right) \end{array}\right.$$
where $S_{mn}(t)$ is the echo signal received by the *n*-th element when the spherical wave with deflection angle $\alpha_{m}$ is transmitted. $S^{CSW}\left(t\right)$ and $S_{m}^{SW}\left(t\right)$ are the imaging functions for the coherent spherical wave and the *m*-th spherical wave, respectively.

By changing the angle of the virtual point source relative to the *z* axis, spherical waves with different deflection angles can be emitted. For each deflection spherical wave, the imaging function can be obtained by beam synthesis according to the second equation of Equation (15), and the reconstructed image of spherical wave emission can be obtained. Finally, K spherical wave images with different deflection directions are synthesized to obtain the imaging results of the composite spherical wave according to the first equation of Equation (15).

In the composite spherical wave imaging algorithm, the imaging range of each deflection emission is larger than that of the plane wave. Therefore, in this sense, the imaging effect of the composite spherical wave is better than that of the composite plane wave.

## 3. Multi Spherical Wave Imaging Method

### 3.1. Multi Spherical Wave Emission

Spherical wave emission in Section 2.2 and Section 2.3 is based on the condition that there is only one kind of wave in the medium. When there are compressional and shear waves propagating in the medium, the transducer can generate multi spherical waves. We hope that the transducer can radiate the spherical compressional wave and spherical shear wave at the same time forming the multi spherical wave emission.

Firstly, we assume that the virtual point source at the spherical center is excited by an electrical pulse, and the compressional and shear waves are simultaneously excited and propagate with different velocity. The signals received by each element of the transducer contain two wave packets corresponding to compressional and shear waves, respectively, from the virtual point source. These signals can be easy calculated and obtained by medium parameters. Then, we load these signals containing two wave packets on the corresponding transducer element for excitation. In this case, each element is excited twice because the excitation signals contain two wave packets.

If the spherical center of the spherical wave is on the *z*-axis, the time delay of the *n*-th element for the first wave packet excitation can be expressed as:(16)$$\Delta t_{{}_{n}}^{p}=T_{c}+\frac{f_{oc}-\sqrt{f_{oc}^{2}+x_{n}{}^{2}}}{c_{p}}$$
where $c_{p}$ represents the compressional wave velocity. Similarly, the time delay of the *n*-th element for the second packet excitation can be expressed as:(17)$$\Delta t_{{}_{n}}^{s}=T_{c}+\frac{f_{oc}-\sqrt{f_{oc}^{2}+x_{n}{}^{2}}}{c_{s}}$$
where $c_{s}$ represents the shear wave velocity.

It can be found that the compressional wave excited by the first packet [under the time delay rule given by Equation (16)] and shear wave excited by the second packet [time delay rule given by Equation (17)] are spherical compressional wave and spherical shear wave, respectively. Therefore, two spherical waves are formed and propagate with compressional and shear wave velocities, respectively. They are equivalent to the spherical compressional wave and spherical shear wave that start simultaneously from the virtual point source, respectively. It should be noted that the shear wave excited by the first packet and the compressional wave excited by the second packet are not the spherical waves and they are not considered here and do not affect the final results.

According to this method, ultrasonic phased array can simultaneously emit spherical waves propagating with the velocities of the compressional and shear waves to scan the measured block, that is, forming multi spherical wave for scanning. Due to the different ultrasonic characteristics of compressional and shear waves, multi-wave imaging with the multi spherical wave emission mode has good influence on the effective detection range and transverse resolution of imaging detection.

### 3.2. Coherent Multi Spherical Wave Compounding

If the spherical center of the spherical wave is not on the *z*-axis, the deflection spherical wave can be obtained. Similar to Figure 6, composite multi spherical wave imaging is performed by changing the deflection angle of the spherical wave in turn. At this sense, the ultrasonic phased array excites spherical waves with different deflection angles and different propagation velocities for different virtual point source positions. For a given deflection angle $\alpha_{m}$ (*m* = 1, 2, ..., *K*), each transducer element needs to be excited twice to form spherical compressional and shear waves, so the whole process of composite multi spherical wave compounding emission has a total of 2*K* excitations. Where, when the deflection angle of the virtual point source is $\alpha_{m}$ (*m* = 1, 2, ..., *K*), the delay time of the first excitation of the *n*-th element can be expressed as:(18)$$\Delta t_{{}_{n}}^{p}=T_{c}+\frac{f_{oc}-\sqrt{{\left(x_{n}-f_{oc}\sin\alpha_{m}\right)}^{2}+{\left(f_{oc}\cos\alpha_{m}\right)}^{2}}}{c_{p}}$$

The delay time of the second excitation of the *n*-th element can be expressed as:(19)$$\Delta t_{{}_{n}}^{s}=T_{c}+\frac{f_{oc}-\sqrt{{\left(x_{n}-f_{oc}\sin\alpha_{m}\right)}^{2}+{\left(f_{oc}\cos\alpha_{m}\right)}^{2}}}{c_{s}}$$

According to the geometric relation in the figure above, when the deflection angle is $\alpha_{m}$, the propagation distance $d_{oc}$ of spherical compressional waves from virtual point to point (*x*, *z*) and then reflected to the *n*-th element of the phased array is the same as that of the spherical shear wave, which can be expressed by Equation (13).

The image function needs to superimpose echo signals of all array elements according to a certain delay. According to Equation (13), time delays of the echo signal received by the *n*-th element are defined as $\tau_{Mm}$ (*M* = 1, 2), which can be written as: (20)$$\begin{array}{cc} \tau_{Mn} & =\frac{d_{oc}-\left(\sqrt{{\left(x+f_{oc}\sin\alpha_{m}\right)}^{2}+{\left(z+f_{oc}\cos\alpha_{m}\right)}^{2}}+\sqrt{x^{2}+z^{2}}\right)}{c}\\ & =\frac{\sqrt{{\left(x-x_{n}\right)}^{2}+z^{2}}-\sqrt{x^{2}+z^{2}}}{c_{M}} \end{array}$$
where $c_{M}$ represents the ultrasonic wave velocity, which is the compressional wave velocity for *M* = 1 and is the shear wave velocity for *M* = 2.

Therefore, coherent multi spherical wave imaging can be obtained as:(21)$$S^{CMSW}\left(t\right)=\sum_{M=1}^{2}\sum_{m=1}^{K}\sum_{n=1}^{N}S_{mn}\left(t-\tau_{Mn}\right)$$

By changing the included angle of the virtual point source relative to the *z* axis, multi wave emission is carried out at *K* different deflection angles. According to Equation (21), all received signals are beam synthesized, and finally, the results of coherent multi spherical wave imaging are obtained.

## 4. Imaging Results and Discussion

In this section, experimental data are used to conduct imaging in accordance with the above methods. The test block in the experiments is shown in Figure 7, and it is made of carbon steel with the wave velocities of $c_{p}$ = 5930 m/s and $c_{s}$ = 3240 m/s. The test area is a group of through-holes distributed in a semicircle. There are 19 circular through-holes with a diameter of 1 mm on the left and 16 circular through-holes with a diameter of 2 mm on the right. The transducer in the experiment is a special transducer designed [17] in previous work for multi-wave detection, which is in direct contact with the test block through an ultrasonic coupling agent. The total number of array elements is 32, and the center frequency is 3.2 MHz.

### 4.1. Undeflected Plane and Spherical Wave Imaging

In the experiments, the single plane wave emission is achieved by non-delay excitation and the single spherical wave is achieved by time delay according to Equation (8). Then, the experimental equipment is used to collect and process the data. Figure 8 shows the sound field snapshots in the process of undeflected plane and spherical wave excitations.

By comparing Figure 8a,b, it can be found that the radiation range of the sound field is consistent with the previous theoretical analysis. The plane wave field is seriously affected by the size of the transducer and its effective scanning range is equivalent to the transducer aperture. However, the radiation range of the spherical wave based on the virtual point source is no longer limited by the transducer aperture.

According to Equations (3) and (11), the echo data collected by the system are calculated for beam synthesis. Imaging results of the single plane wave and spherical wave without deflection can be obtained, as shown in Figure 9 and Figure 10, respectively. The contours shown in Figure 10 are from −1 dB to −3 dB in steps of 0.5 dB relative to the maximum amplitude of the defect echo. Generally, the −3 dB contour is identified as the envelope of the defect shape. In Figure 11, Figure 12, Figure 13 and Figure 14 below, defects are also represented by the contour curves.

By comparing the results in Figure 9 and Figure 10, it can be seen that the through-hole defects in the test block can be detected by both undeflected plane wave imaging and undeflected spherical wave imaging emitted by the virtual point source. Among them, plane wave imaging without deflection can only detect the corresponding defects below the transducer array, while spherical wave imaging without deflection can detect the defects beyond the vertical range of the transducer array. The experimental results of these two methods are in agreement with the theoretical analysis and the sound field model. Similarly, it can be seen that the overall detection effect of the two methods is still insufficient. Since there is no focusing emission, the imaging quality of the single plane wave is relatively low.

### 4.2. Composite Multi Spherical Wave Imaging

In this section, experiments of multi spherical wave imaging and composite multi spherical wave imaging are considered and carried out. First, the transducer controlled the emission of two spherical waves with different wave velocities at 0 deflection angle (without deflection). Then, each element received the defect scattering echo data for beam synthesis according to Equation (21). Where, multi spherical waves are emitted without a deflection angle, image reconstruction of the data after beam synthesis is obtained and shown in Figure 11.

By comparing the results in Figure 10 and Figure 11, it can be seen that the detection sensitivity of ultrasonic multi spherical wave imaging based on virtual point source emission is improved in a wide range of deflection angles, but the overall noise is high. Both imaging methods can detect circular defects with a diameter of 1 mm in a certain deflection direction, but the overall detection effect is poor. This is because spherical wave imaging and multi spherical wave imaging with a single angle also have no focusing emission, and their overall imaging effect is far from that of previous ultrasound focusing imaging, which is also consistent with the theoretical analysis above.

Next, in order to verify the imaging effects of coherent spherical wave compounding and the composite multi spherical wave, the experimental system excited the element according to the delay rules analyzed in Section 2.3.2 and Section 3.2, respectively, so as to launch a series of spherical waves and multi spherical waves with different deflection angles for scanning imaging. Here, for coherent spherical wave compounding and composite multi spherical wave imaging, the phased array transmits spherical wave and multi spherical wave at 11 angles, respectively. In the scanning range of −30°~30°, the array element has been excited once every 6°. Then, according to Equations (15) and (21), the beam synthesis calculation has been carried out according to the defect scattering echo data received by each element, in which *K* = 11. The center of the transducer is selected as the coordinate origin of the imaging results, and the data after beam synthesis is reconstructed. The results are shown in Figure 12 and Figure 13.

Compared with single plane wave imaging, the resolution and signal-to-noise ratio of composite spherical wave imaging and composite multi spherical wave imaging are greatly improved. By comparing the imaging results in Figure 12 and Figure 13, it can be seen that both composite spherical wave and composite multi spherical wave ultrasonic imaging can detect circular defects with diameters of 1 mm and 2 mm within a certain range. However, the number of defects detected by composite multi spherical wave imaging is more in the larger range of deflection angles. Compared with the composite spherical wave, the lateral resolution of composite multi spherical wave imaging is partially improved.

The above results show that the detection result of composite multi spherical wave ultrasonic imaging based on virtual point source emission is better than that of the composite spherical wave. However, the phased array should be excited twice at each deflection angle for the composite multi spherical wave while excited only once for the composite spherical wave. Therefore, the detection time of composite multi spherical wave imaging is slightly slower than that of the composite spherical wave.

In order to further improve the computational speed of composite multi spherical wave imaging, five multi spherical waves with different deflection angles are set up in experiments. That is, in the scanning range of −30°~30°, the value of $\alpha_{m}$ gradually increases, and the tolerance is 15°. Then, each element received defect scattering echo data for the beam synthesis calculation according to Equation (21), in which *K* = 5. The center of the transducer was selected as the coordinate origin of the imaging result, and the data after beam synthesis are reconstructed, as shown in Figure 14.

By comparing the imaging results in Figure 13 and Figure 14, it can be seen that there is little difference in the imaging results when the deflection angle is changed for five or 11 times. It can also be seen that, compared with Figure 12, composite multi spherical wave imaging with the virtual point source changing the deflection position for five times also detects more defects within a larger range of deflection angles on both sides. This indicates that in the same deflection range, composite multi spherical wave imaging can shorten its calculation time by reducing the deflection times of the virtual point source and thus the overall transmission times, and the final imaging effect can also maintain the detection resolution and detection sensitivity of the algorithm at large deflection angles.

In ultrasonic detection and imaging, the transverse and longitudinal resolutions are different. The transverse resolution depends on the effective width of the ultrasonic beam while the longitudinal resolution is mainly determined by the duration of the excitation pulse and the sound speed. Generally speaking, the longitudinal resolution is better than the transverse resolution for ultrasonic array imaging. The defects in the ultrasonic image seem to be oval shaped, not circular. In fact, ultrasonic detection and imaging only proves the existence and approximate size of the defect, not the actual shape of the defect.

## 5. Conclusions

In order to improve the resolution and SNR of plane wave imaging detection in traditional ultrasonic imaging algorithms, the multi spherical wave ultrasonic imaging method based on phased array virtual point source emission has been proposed. In this paper, the flow based on spherical wave emission has been introduced in detail and the calculation formula of echo beamforming for multiple spherical wave emission has been derived. Finally, the imaging method is verified experimentally, and the following conclusions can be drawn:The imaging region of the undeflected plane wave has been seriously affected by the size of the transducer. Plane wave imaging without deflection can only detect the corresponding defects below the transducer array. However, undeflected spherical wave imaging can detect the defects beyond the vertical range of the transducer array.Compared with spherical wave imaging without deflection, the detection sensitivity of the composite multi spherical wave is slightly improved in the sensitivity of the large angle. However, there is no focusing emission, and the overall imaging effect is relatively poor.Compared with undeflected plane wave and spherical wave imaging, the resolution and SNR of composite spherical wave imaging and composite multi spherical wave imaging are greatly improved. Since the properties of both compressional and shear waves are utilized, multi spherical wave imaging can obtain better images over a larger deflection region.Composite multi spherical wave imaging can improve the speed of echo processing by appropriately reducing the number of transmission deflections, and the imaging effect is better than that of composite spherical wave ultrasonic imaging.

## Figures and Tables

**Figure 1 sensors-22-06800-f001:**
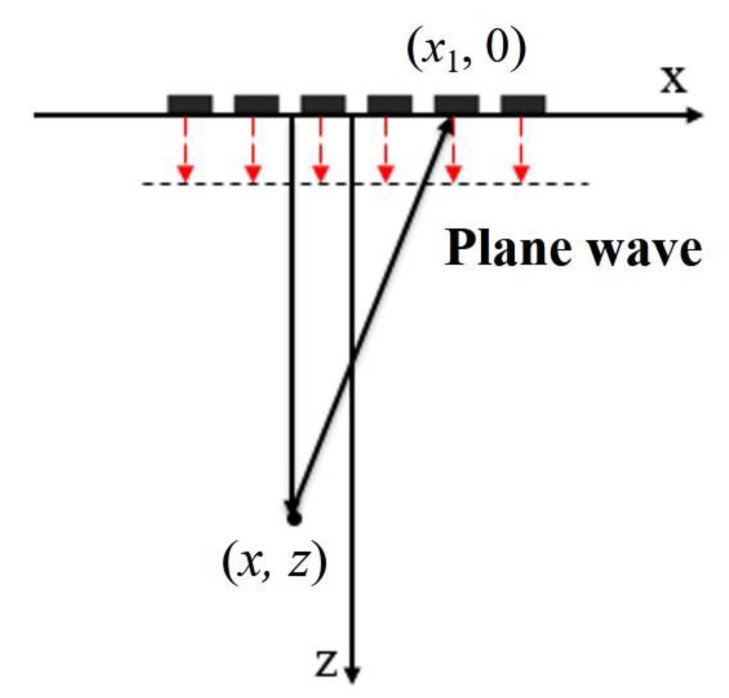
Draft of sound wave propagation in plane wave imaging.

**Figure 2 sensors-22-06800-f002:**
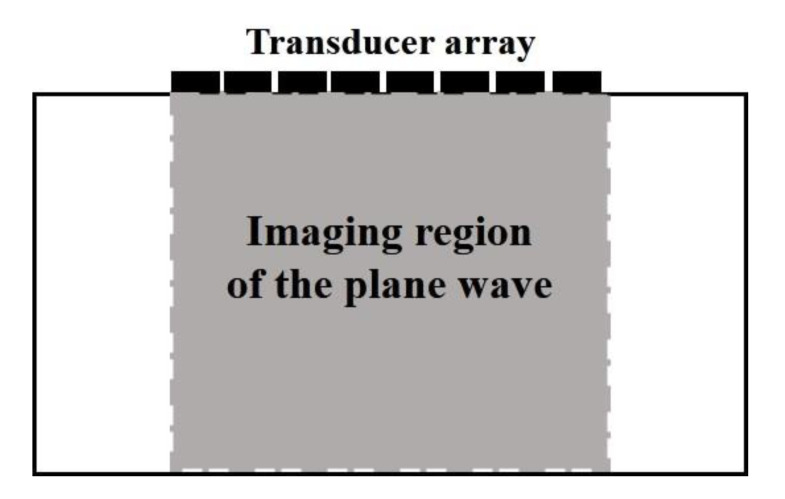
Draft of a single plane wave imaging region.

**Figure 3 sensors-22-06800-f003:**
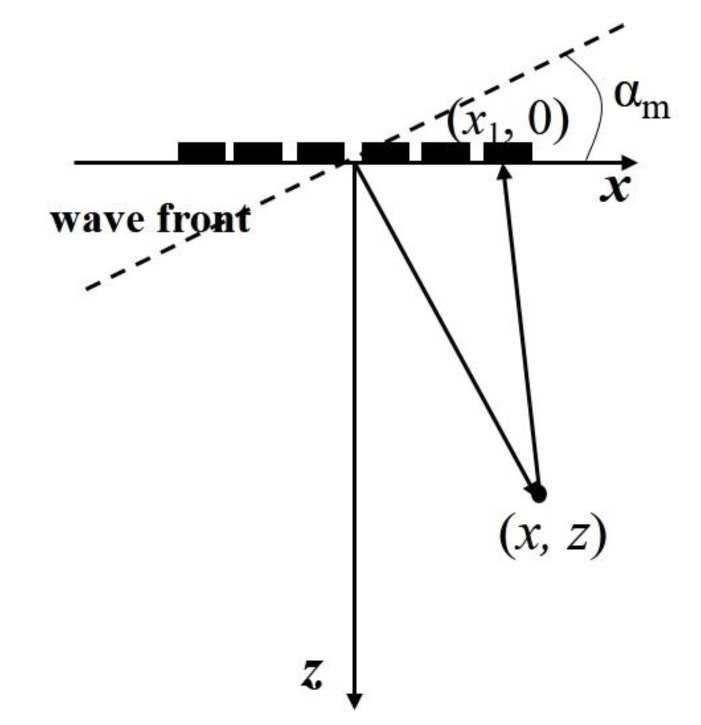
Draft of deflected plane wave emission.

**Figure 4 sensors-22-06800-f004:**
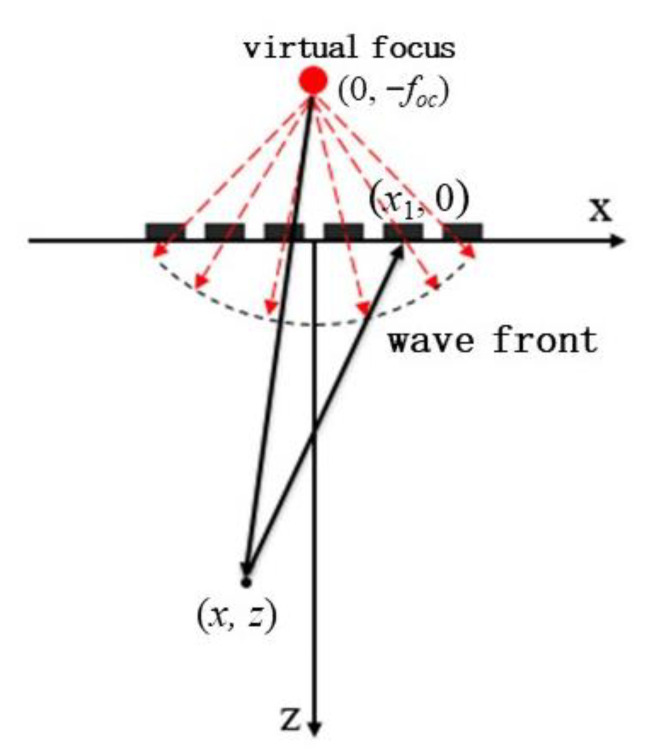
Draft of a transducer array emitting a spherical wave.

**Figure 5 sensors-22-06800-f005:**
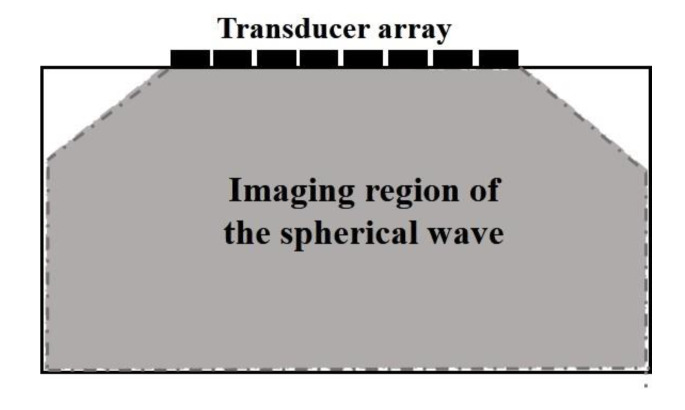
Draft of a single spherical wave imaging region.

**Figure 6 sensors-22-06800-f006:**
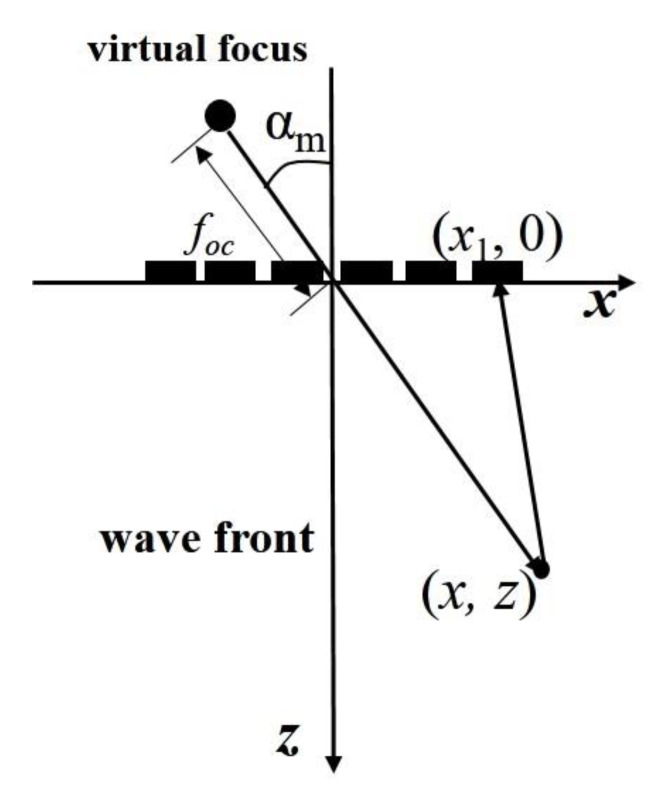
Draft of emission and propagation of the deflected spherical wave.

**Figure 7 sensors-22-06800-f007:**
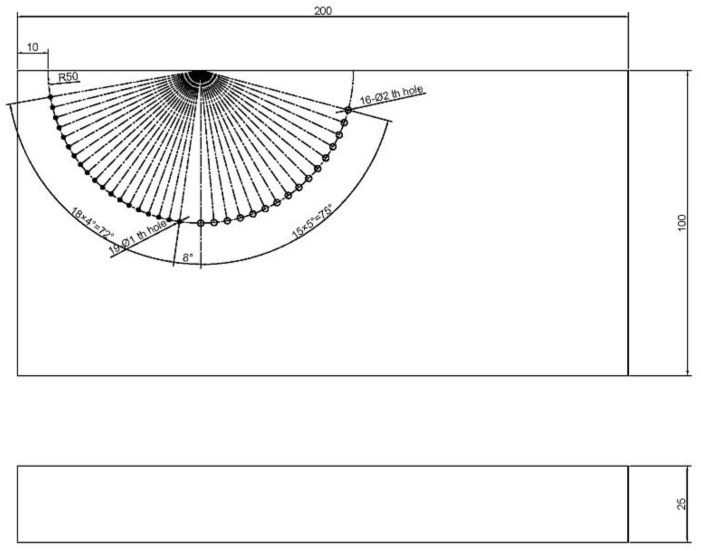
Draft of the test block.

**Figure 8 sensors-22-06800-f008:**
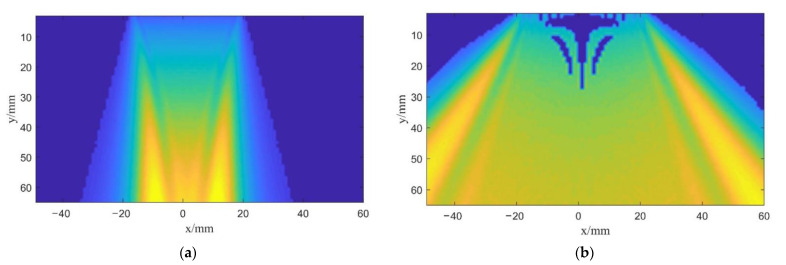
Sound field snapshot emitted by: (**a**) A single plane wave without deflection; (**b**) A single spherical wave without deflection.

**Figure 9 sensors-22-06800-f009:**
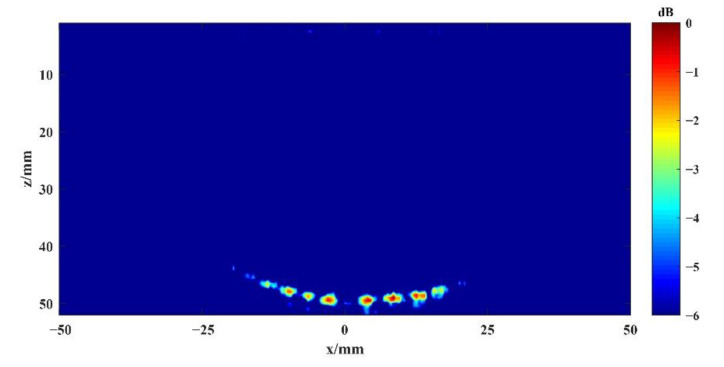
Schematic diagram of the undeflected plane wave imaging result.

**Figure 10 sensors-22-06800-f010:**
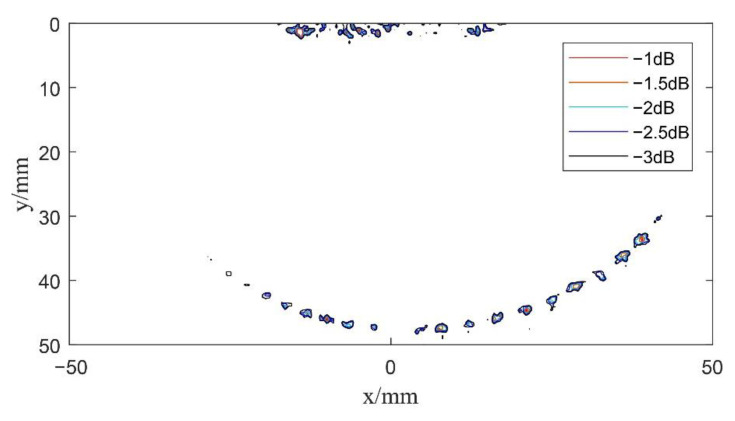
Schematic diagram of the undeflected spherical wave imaging result.

**Figure 11 sensors-22-06800-f011:**
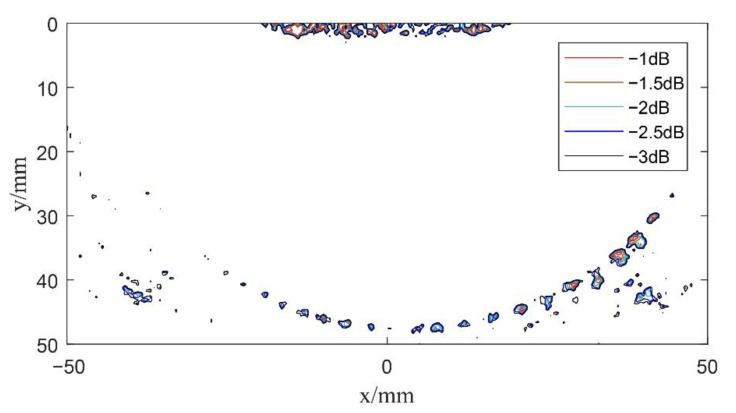
Schematic diagram of multi spherical wave imaging results without deflection.

**Figure 12 sensors-22-06800-f012:**
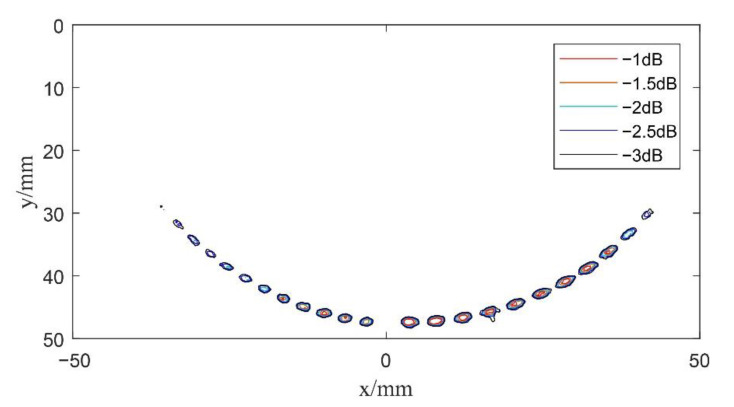
Schematic diagram of composite spherical wave imaging (*K* = 11).

**Figure 13 sensors-22-06800-f013:**
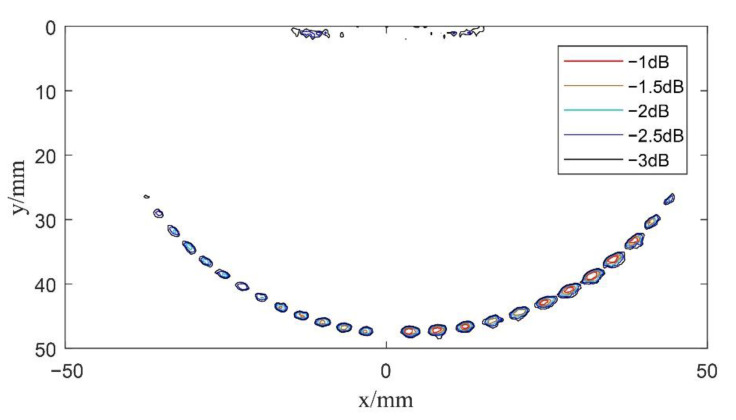
Schematic diagram of composite multi spherical wave imaging (*K* = 11).

**Figure 14 sensors-22-06800-f014:**
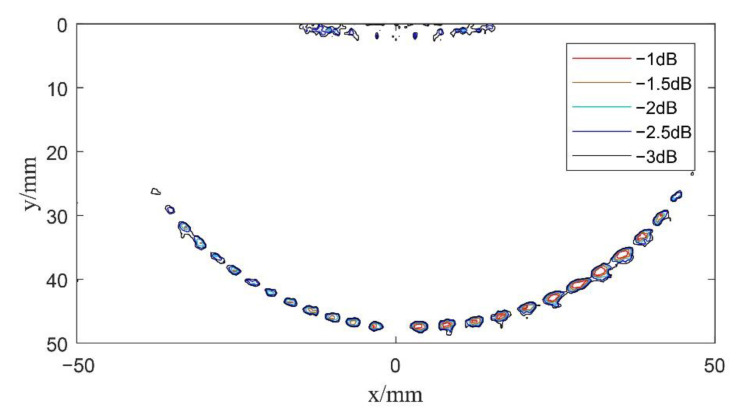
Schematic diagram of composite multi spherical wave imaging (*K* = 5).

## Data Availability

Not applicable.

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
