# Peer review of "Multi Spherical Wave Imaging Method Based on Ultrasonic Array"

_sensors, 2022, doi:10.3390/s22186800_

Round 1

Reviewer 1 Report

Authors proposed multi spherical wave imaging methods based on ultrasound array. Mathematical analysis and simulation results for proposed methods are very good and effective to show the advantages. English grammar looks fine. According to reviewer’s opinion, the manuscript could be minor revision and be published with some suggestive comments.

1. Please use abbreviated journal names in the reference section.

2. Please provide the city and country information of the proceeding papers in the reference section.

3. Please describe the difference between phase array imaging and your proposed method.

4. Authors mentioned that "the imaging resolution and SNR are not improved" in Line 186. Is there any reason ?

5. What is the K deflection angles in Line 196 in Figure 6 ?

6.  Which kinds of the software authors used ? It looks like Field II. If so, please describe the condition of the simulation for Figure 8 such as number of the arrays and operating frequency.

7.Please provide the ref. for the sentence (At present, with the improvement of testing~) with the ref. (https://journals.plos.org/plosone/article?id=10.1371/journal.pone.0249034)

8. In Conclusion section, authors might insert some important simulated performances.

Reviewer 2 Report

This paper proposes a very interesting wave imaging method to be used with ultrasonic arrays based on a spherical wave approach. This method has a big potential to improve detection range in comparison with traditional plane wave approaches. This hypothesis seems to be proved in the experiments carried out by the authors.

I think this paper is an excellent one and has to be published. Some minor details are, in my humble opinion:

1. Reference to Figure 8 in line 284 could be changed to Figure 7. Also one dimension of the test block is omitted in the figure (100 and 200mm in x and y axes, but not in the z-axis). This is important for reproducibility of the proposal.

2. Results of spherical wave imaging showed in Figures 10-14 could be better if the target holes could be shown in the figures as overlapped contours. Figures seems to be properly scaled (twice width than height, corresponding with double size) but holes seems to be with oval shape, not circular. It could be a visual effect with the figures or a distortion in detected shapes?
